# Professional stakeholders' expectations for the future of community pharmacy practice in England: a qualitative study

Evgenia Paloumpi,[1] Piotr Ozieranski [iD],[2] Margaret C Watson [iD],[3] Matthew D Jones [iD] [1]

[1]Department of Life Sciences, University of Bath, Bath, UK
[2]Department of Social & Policy Sciences, University of Bath, Bath, UK
[3]Strathclyde Institute of Pharmacy and Biomedical Sciences, University of Strathclyde, Glasgow, UK

**Correspondence to**
Dr Matthew D Jones;
m.d.jones@bath.ac.uk

## ABSTRACT

**Objectives** To explore the views of professional stakeholders on the future of community pharmacy services in England. Specific objectives related to expectations of how community pharmacy services will be provided by 2030 and factors that will influence this.

**Design** Qualitative, using semistructured interviews in person or via telephone/Skype. The topic guide was informed by a recent policy review that used the Walt and Gilson policy framework. Transcripts were analysed using inductive thematic analysis.

**Setting** England.

**Participants** External stakeholders were representatives of non-pharmacy organisations, including policy-makers, commissioners and representatives of healthcare professions. Internal stakeholders were community pharmacists or pharmacy organisation representatives. Interviewees were identified using stakeholder mapping

**Results** In total, 25 interviews were completed (7 external stakeholders and 18 internal stakeholders, of which 10 were community pharmacists). Community pharmacy was recognised as having a key role in expanding health system capacity ('…*pharmacy is the obvious person to take on those roles…*'), particularly for long-term condition management (eg, adherence, reducing polypharmacy, monitoring), urgent care (eg, minor illnesses) and public health (including mental health). For these contributions to be developed and optimised, greater integration and collaboration with general practices will be needed ('…*there is no room for isolationism in pharmacy anymore…*'), as well as use of technology in a patient-centred way and full access to health records. These changes will require workforce development together with appropriate commissioning and contractual arrangements. Community pharmacy is currently undervalued ('…*the complete misunderstanding by senior Government officials is very scary*') and recent investment in general practice pharmacists rather than community pharmacy was seen as a missed opportunity.

**Conclusions** Community pharmacy as a sector could and should be developed to increase health service capacity to address its current challenges. Numerous modifications are required from a range of stakeholders to create the environment in which these changes can occur.

## STRENGTHS AND LIMITATIONS OF THIS STUDY

⇒ The key strength of this study is its examination of the development of community pharmacy services in England in their entirety (rather than individual services) from the perspective of a wide variety of systematically selected interviewees, making it more representative and generalisable than some previous studies.

⇒ The inability to recruit stakeholders from the government and organisations that represent the public limits the breadth of perspectives included.

⇒ The addition of pharmacists working in other sectors (eg, general practitioners pharmacists) could have added useful insights.

## INTRODUCTION

The National Health Service (NHS) in England (the largest and most populous part of the UK) is currently facing unprecedented work pressures. In primary care, increased patient demand combined with a reduction in the number of general practitioners (GPs), poor morale and decreased real-terms funding per capita has led many to talk of a system in 'crisis'.[1] The secondary care sector is facing similar challenges.[2]

Community pharmacies are one of four fundamental sectors of primary care in England.[3] They are easily accessible premises where qualified healthcare professionals supply medicines and provide medicines-related and public health services. There is evidence of the effectiveness of community pharmacy services to manage long-term conditions,[4] minor ailments[5] and promote public health[6] from systematic reviews and quantitative primary research. Greater use of community pharmacy has, therefore, been identified as an important way to address current NHS challenges.[7] However, despite the substantial contribution that community pharmacies make to the NHS throughout England, particularly during the COVID-19 pandemic, the sector is experiencing sustained underfunding,[8] increased rates of closure[9] and growing competition from online pharmacies.[10]

In addition, since 2015 there has been a large increase in the number of 'general practice pharmacists' in England.[11] This is a distinct role to that of community pharmacists, responsible for various tasks in general practices, such as structured medication reviews and medicines reconciliation. The impact of GP pharmacists on community pharmacy has not been widely explored.

A tipping point is being reached for community pharmacy in England, with an urgent need for a national vision and strategy.[12] A multistakeholder perspective combined with a system-wide approach is needed to ensure that strategic developments reflect not only need but capabilities. There has been no recent exploration of stakeholders' visions for the future of the community pharmacy and earlier consultations have focused on singular activities.

This study explored the views of health-sector stakeholders on the short-to-midterm (10-year) future of community pharmacy in England. Specific objectives were to explore:

► Expectations of how community pharmacy services will be provided by 2030.
► Factors that will influence future service development and delivery.
► Opinions of how community pharmacy services should ideally be provided.
► Facilitators and barriers to the development of these ideal community pharmacy services.

## METHODS
A qualitative interview design was chosen to allow in-depth exploration of stakeholders' views. Reporting is in accordance with the Consolidated Criteria for Reporting Qualitative Research (online supplemental table S1).

### Interviewees and recruitment
Both internal and external stakeholders were recruited. External stakeholders were representatives of non-pharmacy organisations (online supplemental table S2). Internal stakeholders were either current community pharmacists or representatives of pharmacy organisations. Internal and external organisational representatives were recruited by stakeholder mapping and direct invitation, whereas community pharmacists were recruited via advertisement. Interviewees gave written or audio informed consent.

Stakeholder mapping was used to systematically identify and invite a broad range of internal and external organisational representatives (such as policy-makers, commissioners and representatives of patients or healthcare professionals) to participate.[13] An initial list of stakeholders was constructed using a policy review[14] and research team expertise (online supplemental table S2). A snowballing approach was also adopted to supplement any underrepresented stakeholder categories. Stakeholder profiles (career history, position, interest and influence) were created using publicly accessible information. These were used to select stakeholders to invite to interview. Up to five stakeholders were identified and ranked within each organisation. The highest ranked stakeholder was sent a written invitation, with telephone follow-up if needed. If a stakeholder declined, the next ranked individual from their organisation was invited. This process continued until a representative of every organisation was interviewed or all the identified stakeholders from an organisation had declined their invitation.

Community pharmacists were eligible for inclusion if they worked regularly in a community pharmacy in England and were identified using purposive sampling, to obtain a diverse sample based on a sampling matrix including type of community pharmacy, locum status (as these pharmacists work in multiple pharmacies and so may have different perspectives), gender and geographical region. The study was advertised to community pharmacists via the Royal Pharmaceutical Society's (ie, the professional body for pharmacy) online forums, local pharmaceutical committees, the research team's networks and social media. Every eligible person who responded to these advertisements was interviewed.

### Data collection
Interviews were carried out by one trained researcher (EP) face to face, by telephone or Skype, to provide flexibility and increase recruitment.[15] A topic guide (online supplemental material) was developed based on a recent policy review[14] that employed the Walt and Gilson policy framework[16] and discussion within the research team. One pilot interview was conducted. Interviews were audio-recorded, transcribed verbatim and checked for accuracy and anonymity. The interviewer made notes of key points and observations from the interview.

### Data analysis
Interview transcripts were analysed using inductive thematic analysis, which is suitable for applied health research aiming to inform the development of policy.[17] Analysis was iterative, based on a six-phase process: data familiarisation, generating initial codes, developing themes, reviewing themes, defining and naming themes and writing the analysis.[17] Initial manual coding was then interpreted to generate initial themes and subthemes using NVivo V.12 software. Themes were iteratively refined for cohesiveness using a thematic map, before the final themes were named and defined. These processes were led by one researcher (EP), with regular discussion and agreement with the team.

### Patient and public involvement statement
Patients and the public were not involved in this study. A separate study examining the perspectives of members of the public has been completed and will be reported.[18]

## RESULTS
### Interviewee characteristics
In total, 46 stakeholders were invited and 25 interviews were completed between November 2018 and April 2019.

**Table 1** Themes and subthemes generated from the analysis

| Theme | Subtheme |
|---|---|
| Creating collaborations for wider pharmacy services provision from community pharmacy | Collaboration: expected, ideal and a facilitator |
| | Existing relationships with general practices |
| Integrating new technology into pharmacy services | Technology expected to change community pharmacy practice |
| | Making technology work for patients |
| | Information-sharing technology |
| Building workforce competency | Increasing responsibilities with a decreasing workforce |
| | Enhancing the skills of the workforce |
| | Role of pharmacy technicians |
| Future community pharmacy services | Community pharmacists' role in managing chronic conditions, urgent care and prevention |
| | Opportunities for further community pharmacy development |
| | Creating the environment for future developments |

There were 7 external stakeholders from 7 organisations and 18 internal stakeholders. Of these, 8 were organisational representatives (from 7 organisations) and 10 were community pharmacists (online supplemental tables S3 and S4), (supplementary material). Stakeholders representing three patient organisations, two state policy-making organisations, two pharmacy representative organisations and three organisations representing other healthcare professions were invited, but did not participate. Interview duration varied from 35 to 75 min.

### Overview of themes

Four themes with 11 subthemes were generated from the analysis (table 1). Each of these will be discussed in order of dominance.

There were large areas of convergence in the views discussed by external stakeholders and internal stakeholders (both organisational representatives and community pharmacists) (figure 1). In particular, they related to community pharmacy services, technology and the workforce. There was also considerable divergence with only community pharmacists discussing how the sector felt insecure and undervalued and expressing concerns about technology and collaborative relationships.

### Theme 1: collaboration for wider pharmacy services provision from community pharmacy

#### Subtheme: collaboration: expected, ideal and a facilitator

There is a coming together of health and social care…and I think pharmacy has to be in that tent…I

don't see any future…for a pharmacy as a kind of standalone… there is no room for isolationism in pharmacy anymore it has to be seen as part of a wider team. Interviewee ID12—external stakeholder (GP)

Interviewees expected future community pharmacy services to be provided collaboratively with other healthcare professionals and voluntary organisations. They envisaged more integration between professions with greater awareness of other professionals' skills and an understanding that services could be provided differently. Interviewees expected community pharmacists to work within local groups of general practices (known as primary care networks in England) and viewed current involvement as an encouraging step towards integration into primary care. Interviewees also described community pharmacists' strategic involvement in local and national structures as part of an ideal model of care. They emphasised that a community pharmacy culture change that promotes collaborative working would be crucial for patient care and the sector's survival.

#### Subtheme: existing relationships with general practices

…patients speak the language of medicines in terms of 'the blue ones' or 'the pink ones'… community pharmacists…straddle a divide between how patients think and talk and how the system thinks and talks…I deal with medicines every day of the week but somebody who is sat at their desk [GP pharmacist] wouldn't have a clue where they start with that. Interviewee ID14—community pharmacist

There were contrasting opinions about the role of GP pharmacists in supporting community pharmacy services: external stakeholders viewed GP pharmacists as facilitators, but community pharmacists referred to them as a barrier. Internal stakeholders were disappointed that NHS England had invested mostly in GP pharmacists rather than the existing community pharmacy workforce, describing this as a missed opportunity. There was concern over a division between 'clinical' GP pharmacists and 'community' pharmacists, and some thought that community pharmacists were better placed to support patients, due to better practical knowledge of patients' current medicines and how they are using them.

You only need to look at the debacle with the flu service…both GPs and pharmacists think they are in competition with each other and rather than working collaboratively…for the whole population… Interviewee ID10—community pharmacist

Many community pharmacists described competitive 'them and us' (Interviewee ID20) relationships with GPs, resulting from conflicting financial incentives (eg, competition between community pharmacies and GP practices to vaccinate the same pool of patients against influenza) and poor understanding of pharmacists' skills (eg, a reluctance by some GPs to share health records due to the view that pharmacists are not trained to protect confidential information). However, some believed that

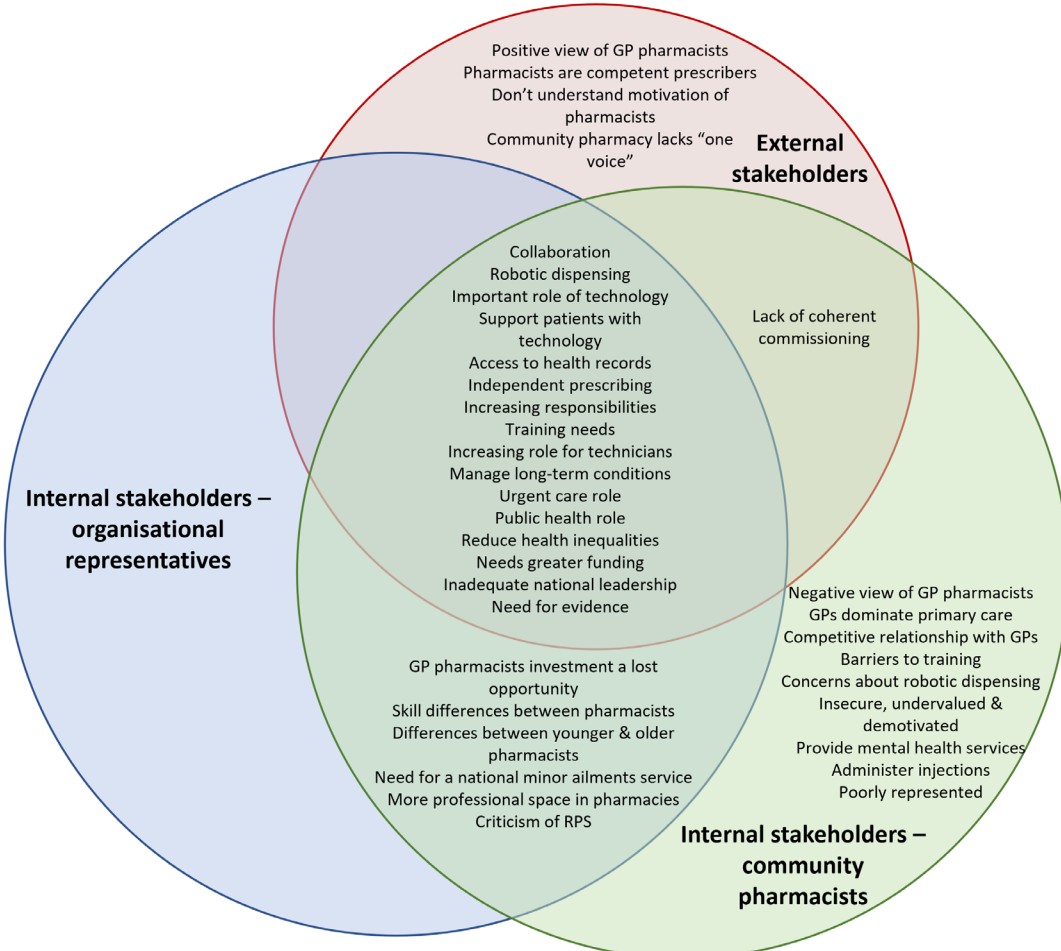

**Figure 1** Areas of convergence and divergence in the views discussed by external stakeholders and internal stakeholders (both organisational representatives and community pharmacists). GP, general practitioner; RPS, Royal Pharmaceutical Society.

GPs valued community pharmacists and were optimistic about future relationships.

Community pharmacists perceived GPs as dominating primary care, mentioning 'lack of parity' (Interviewee ID14) in the distribution of services and highlighting the need to align the medical and pharmaceutical NHS contracts. Overall, the GP contract and workload pressures in general practice were the most influential factors driving pharmacy service development, as the capabilities of community pharmacy were only discussed by a minority of interviewees.

### Theme 2: integrating technology into pharmacy services
#### Subtheme: technology expected to change community pharmacy practice

If we can use IT robotics just to take care of certain aspects which are more tedious and laborious…the pharmacists will have much more time to devote to patient care. Interviewee ID15—external stakeholder

New technology was seen as integral to future pharmacy services and healthcare in general. This was attributed to patient expectations, the entry of technology companies into the medicines market and the government's intentions.

Interviewees described a 'hub-and-spoke' model, with automated dispensing of medicines in a central location, away from community pharmacy premises. This was expected and perceived as ideal, and was usually viewed as enabling a change from a medicine supply model to one based on an enhanced clinical role. However, community pharmacists expressed concerns about delays to patient care and increased costs.

Home deliveries of medicines and increasing use of online pharmacies were commonly connected and were expected to increase, driven by convenience. Use of artificial intelligence to support patient decision-making, accurate prescribing and to reduce dispensing errors was also discussed speculatively, mainly by external stakeholders.

… by 2030 Amazon could be delivering peoples' prescriptions door-to-door so where does pharmacy fit, if pharmacy hasn't got a more defined and a more developed clinical care role then, where is pharmacy? Interviewee ID12—external stakeholder

Many highlighted the impact that these developments might have on the role of community pharmacists. Due to the risk of large wholesale companies dominating the medicines supply market, it will be necessary to establish

a role based on supporting patients. A systematised, transactional and supply-only model for community pharmacy lacking knowledge of regular customers would have adverse consequences for patients.

### Subtheme: making technology work for patients

…wearable technology…clinicians to deal remotely with the way that you are feeling…the fact that you are unwell can be detected at distance and then dealt with either at distance or…face-to-face… Interviewee ID4—internal stakeholder

But it's how patients use that medicine and how they become effective with the medicine that is going to require the clinical expertise of the community pharmacist and that clinical expertise has to be done really face-to-face so that you can see the patient more effectively. Interviewee ID10—community pharmacist

All interviewees described the growing role that healthcare applications (apps) and devices would have in pharmacy services, expecting transformed patient interaction for the supply of medicines, diagnosis and review of patient data. Future patients will require community pharmacists to understand and assist with these sources of information. Interviewees expected increased use of online consultations driven by patient expectations and preference but highlighted the importance of maintaining face-to-face contact with patients. Community pharmacists expressed concerns that face-to-face consultations were more effective. A 'mixed model' of digital and face-to-face communication was seen as ideal.

…we have to be careful we don't generate technology poverty, so we don't want to exclude a sector of the population. Interviewee ID23—internal stakeholder

Interviewees were cautious about how technology might exclude older generations. Community pharmacists were considered fundamental for preventing such inequalities and promoting familiarity with technology.

### Subtheme: information-sharing technology

… I think once that link [read and write access to health records] is…established…in terms of what we can then do…it opens it up…anything and everything really. Interviewee ID21—community pharmacist

All interviewees referred to community pharmacist access to health records as a crucial facilitator for pharmacy services and for some community pharmacists this was ideal. It would allow seamless care, enable community pharmacists to manage long-term conditions more effectively and relieve GP workload. Current read-only access to Summary Care Records (an electronic summary of patients' key clinical information) was beneficial, but the ability to add information is required. Some interviewees thought this would enable community pharmacists to communicate medication changes, thus ensuring their advice is followed and providing GPs with an accurate picture of their interventions.

The personalised care and the genomics is something that…pharmacists…need to make themselves really familiar with because that is the future… Interviewee ID18—internal stakeholder

Interviewees expected new sources of patient data from developments such as pharmacogenomics. Professionals will need to understand and interpret these data, for which community pharmacists should be prepared.

### Theme 3: building workforce competence
### Subtheme: increasing responsibilities with a decreasing workforce

The current Government agenda in relation to the closure of high street pharmacy, …I think that was scary…I have had a very senior person say to me we need to close these pharmacies because all they do is sell shampoo. I think the complete misunderstanding by senior Government officials is very scary. Interviewee ID8—external stakeholder

Most community pharmacists expressed insecurity about their future and felt their value was not recognised due to funding cuts and policies aiming to reduce the number of pharmacies. Lack of career development and incentives for community pharmacists was another factor contributing to a declining workforce. Community pharmacists referred to a tired, disillusioned and demotivated workforce working in tough conditions, warning that many might choose alternative careers (eg, GP pharmacy), thus further reducing the workforce.

Nevertheless, interviewees also expected that community pharmacy responsibilities will increase, referring to community pharmacies becoming an increasingly clinical environment with a reduced medicines supply function. Some community pharmacists expected a transfer of services from GP practices. Therefore, interviewees envisaged that in 2030 there would be more than one pharmacist per pharmacy. For some, this was ideal, providing time for patient interaction and clinical roles.

### Subtheme: enhancing the skills of the workforce

It is a no brainer to me that all pharmacists should be independent prescribers as an absolute pre-requisite…I think the independent prescribing as it currently stands is patchy, it's not many community pharmacists doing it and it is not joined up, it is not really getting us anywhere. Interviewee ID12—external stakeholder

I think at the moment the training for pharmacists working in community pharmacy is very piecemeal and there is…the problem of…getting released from work… Interviewee ID3—internal stakeholder

It was clear that more community pharmacists should become independent prescribers with appropriate funding and training support. Prescribing was a facilitator that would allow community pharmacists to improve long-term condition management, run clinics for specific diseases and release more GP time for complex cases. External stakeholders supported the competence of pharmacists for this role.

Interviewees referred to the need to train the community pharmacy workforce for such responsibilities, including physical examinations, clinical assessment, consultation skills and data management. Many internal stakeholders believed there were skill differences in these areas between generations of pharmacists. Training alongside other healthcare professionals, particularly doctors, will improve mutual understanding. Most community pharmacists described significant barriers to training, including time, funding and accessibility. They believed government expectations were unrealistic in the absence of protected time or funding, highlighting the need for more resources.

> Some pharmacists are their own worst enemy…a lot of them want to hold onto the dispensing process… they're quite happy with how things are… Interviewee ID1—community pharmacist

Interviewees also referred to the sector's resistance to change as one of the biggest barriers to pharmacy services, especially those who might prefer the current situation or be reluctant to risk change to a profitable business. Some stakeholders believed that the businesses that were unwilling to change would suffer. Consequently, some external stakeholders highlighted the importance of understanding community pharmacists' values and motivations. Some internal stakeholders perceived younger pharmacists as advocating for a clinical role, with the contrasting dispensing-focus of some older pharmacists seen as inconsiderate of their legacy to future generations.

### Subtheme: role of pharmacy technicians

> …there are massively differing standards of technicians. There are some that I would trust to run a pharmacy with my life and there are others that I would not let near a pharmacy with a bargepole… Interviewee ID21—community pharmacist

Interviewees recognised pharmacy technicians as underused and suggested making greater use of their skills. A variety of roles were suggested, such as enhancing their role in medication supply to increase cost-effectiveness and release more clinical time for pharmacists. Additional roles included simple consultations, smoking cessation, weight management, influenza vaccination and domiciliary visits. Training will be essential, considering the existing variability in the skills of pharmacy technicians described by some community pharmacists. However, funding for such training was a concern for some community pharmacists, as recent funding cuts had impacted on investment in technician development.

### Theme 4: future community pharmacy services
### Subtheme: community pharmacists' role in managing long-term conditions, urgent care and prevention

> These [long-term conditions]…cannot be dealt with by general practice alone and therefore pharmacy is the obvious person to come on and take on those roles so I can see that aspect growing significantly. Interviewee ID20—external stakeholder

Interviewees expected that community pharmacists' role in the management of long-term conditions would be enhanced in the future, to reduce medicine-related hospital admissions and release GP time. Community pharmacist support of medication adherence was already beneficial and there are opportunities for improving medicines safety, polypharmacy, monitoring and screening services. Drivers for these developments included current success, an existing team of capable staff, and rising multimorbidity, life expectancy and polypharmacy. Some community pharmacists viewed a more prominent role in long-term condition management as ideal, envisaging a future where they would be adequately resourced to monitor and manage medicines for long-term conditions such as asthma, diabetes and cardiovascular disease. However, some interviewees did not foresee further developments for community pharmacy in long-term condition management, with one warning of a 'difficult journey' (interviewee ID12—external stakeholder) requiring collaborations with other healthcare professionals that is difficult for pharmacists who are 'tied to a pharmacy' (interviewee ID9—internal stakeholder) with limited capacity to take on additional roles and access training. This subtheme, therefore, links to others related to collaboration, the potential of technology to change pharmacy practice and workforce competence.

> There would need to be a change in the…training that a pharmacist gets where it becomes more of a hybrid between being a pharmacist and being a doctor-physician's assistant…without that additional clinic support it's difficult to see how emergency services offered through community pharmacies would grow much beyond the current minor ailment-type model. Interviewee ID20—external stakeholder

Further developments in urgent and emergency care were also expected. Interviewees identified this as an area where the community pharmacist's role has progressed and will continue to develop quickly. Examples included in the NHS 111 triage system, the NHS Urgent Medicine Supply Advanced Service and encouraging local services. Community pharmacy will be involved in reducing visits to emergency departments and thus reducing costs. This was seen as one of the most likely developments, due to its strong presence in national policies. However, interviewees did not view community pharmacies as emergency drop-in centres, as they can only respond to a specific set of conditions.

Community pharmacies were described as the first port of call for minor illnesses, such as skin complaints and eye infections. However, some internal stakeholders believed that community pharmacies could do more with a nationally commissioned minor ailments service. Independent prescribing was also a facilitator for greater involvement in this area. Providing care for minor ailments was perceived as limiting community pharmacies' ability to provide more advanced services. In addition, interviewees

of all types questioned the competence of some pharmacists to provide advanced urgent care.

> I would expect to see definite growth…as much as NHS England talks about the prevention agenda they'll start to engage with community pharmacy which is the largest single healthcare provider of both patients and people before they get ill Interviewee ID14—community pharmacist

Community pharmacies were also expected to have a prominent future role in promoting wellness and disease prevention, because of existing public health services, accessibility, well-equipped pharmacy teams and the advantage of providing care to healthy individuals. However, it was thought that community pharmacy would require improved remuneration for this role.

### Subtheme: opportunities for further community pharmacy development

> …pharmacies become health hubs in the future and if they did…they need to be bigger than they are today…they need to have a different look, feel, access information. Interviewee ID9—internal stakeholder

There was a belief that community pharmacies were well located to serve the needs of their communities and should ideally become centres for advice and health support in one location. However, some internal stakeholders suggested that pharmacies need more space, with less emphasis on retail and a more professional environment.

> My biggest one [hope] would be that community pharmacy would be recognised as a part of a social solution and not just a clinical solution and would be able to sell itself…within an integrated care system… thinking about what…social value could be exploited out of community pharmacy in terms of people, place and planet. Interviewee ID8—external stakeholder

Interviewees envisaged a future where community pharmacy's role included tackling health inequalities, including ensuring that care was accessible in socially deprived areas. This might involve a system of 'prescriptions' for pharmacy services and involvement in social prescribing schemes.

> Absolutely there is opportunity for us to help people with mental health needs both from a chronic and low level to the more acute; there's definitely stuff that community pharmacy can do. Interviewee ID5—internal stakeholder

Mental healthcare in community pharmacies was presented as an opportunity, due to increasing demand and national recognition. Some community pharmacists described mental health services that were currently provided included advice, reassurance and signposting. Community pharmacies were seen as advantageous for providing mental healthcare due to their convenient location, local knowledge and patient relationships that

could facilitate early detection of symptoms and initial assessment. Interviewees proposed a range of interventions that might be provided, including signposting, lifestyle advice, medication adherence monitoring, adverse effect management and provision of specialist services such as Cognitive Behavioural Therapy in consultation rooms. Some strongly believed that antidepressants should be included in the NHS New Medicines Service, describing their exclusion (at the time of data collection) as 'appalling' (ID23). However, some interviewees described mental health services as the least likely development, due to lack of resources and training needs.

Community pharmacists described the success of influenza vaccinations in community pharmacies, which are convenient and reduce GP workload. Based on this, interviewees described opportunities to expand community pharmacists' role in vaccination and administering injections including denosumab, vitamin $B_{12}$, antipsychotic depot injections and corticosteroids, although others were reluctant to see this.

### Subtheme: creating the environment for future developments

> We are not really too concerned about the level of care that we give to our patient and again that purely goes down to the way we are funded…if we were to pay pharmacies more for hitting quality targets, for looking after patients…I think then we would see better care… Interviewee ID22—community pharmacist

Interviewees could not envisage an ideal pharmacy services model without greater funding. They argued that reimbursement based on dispensing volume instead of service quality was not fit for purpose and hindered the community pharmacy workforce in using its clinical expertise to fully benefit patients. Some community pharmacists also expressed disappointment at providing clinical services without being reimbursed.

> I think one of the challenges for pharmacy is consistency and continuity…if we've got lots of disparate offerings from lots of different pharmacies that confuses patients, whereas what I want to see is a more standardised nationalised commissioning of services. Interviewee ID23—internal stakeholder

The need for support from commissioners at all levels was acknowledged. Some community pharmacists described feeling lost due to the lack of a coherent approach to commissioning, referring to NHS England's 'Community Pharmacy Clinical Services Review' as abandoned and to unfulfilled plans for newly commissioned services. They believed that long-term plans would allow investment, highlighting the influence of local commissioners and the importance of consistency.

External stakeholders from a commissioning background stated that a coherent commissioning model was one of the biggest challenges for pharmacy services, highlighting the current model's complexity. Commissioning for community pharmacy was described as fragmented

compared with commissioning of other primary care providers and more integrated thinking was required.

> Improve our image with the community…because if you ask a patient 'well what do pharmacists do?' they just think we pop up in the back of a pharmacy, wear lab coats and hand them out…this is one of the reasons why other healthcare professionals don't even trust us as much as they should. It is because our professional body hasn't done enough to demystify our skills. Interviewee ID22—community pharmacist

Many community pharmacists felt dissatisfied with their representation. They referred to the lack of a 'true voice for pharmacists' (ID14) in influential bodies, at every level of the NHS, and to other healthcare professionals and the public.

> … in my 13 years as a qualified pharmacist, I have not seen anything from either NHS England, the Chief Pharmaceutical Officer, the Department of Health and Social Care or Her Majesty's Government, I have seen no action that actually backs up their nice words for community pharmacy. I don't see that they trust community pharmacy to do anything other than simply dole out medicines. In fact, that is the precise term that NHS England Chief Executive said, why are we spending all this money doling out medicines which was really quite an offensive thing to say but that is all the NHS sees us for. Interviewee ID24—community pharmacist

It was important to have pharmacists working in strategic positions in clinical commissioning groups (since replaced by integrated care boards) and the Department of Health and Social Care to drive future changes. Support from other representatives such as the Royal Pharmaceutical Society, Pharmaceutical Services Negotiating Committee and Chief Pharmaceutical Officer was considered essential. Interviewees referred to inadequate national support and leadership for community pharmacy. Many internal stakeholders criticised the Royal Pharmaceutical Society as a representative body, stating that it did not advocate beyond medicines supply and questioned whether it would be fit for purpose in the future. Some external stakeholders highlighted the need for the pharmacy profession to speak with one voice. All of this led to poor recognition of the sector by the state. Interviewees with policy-making experience were still unaware of the role of community pharmacy and highlighted the need to define the role of community pharmacy in policy. There was a disconnect between the governmental and community pharmacy view of the sector's strengths, with the need for the government to trust community pharmacy and see it as 'integral'.

Evidence was considered an important factor that would contribute to better recognition for community pharmacy. Interviewees from all backgrounds acknowledged the importance of strengthening the evidence base for community pharmacist interventions through enhanced research and intervention recording. Creating evidence responsive to NHS challenges was significant. An external stakeholder with commissioning experience referred to the need for 'iterative', 'applicable at a wider scale', 'affordable' and 'timely' evidence (ID17), while another identified a 'huge research gap' (ID8) for community pharmacy interventions with an impact on social care.

Some suggested that all community pharmacy contractors work together to do this, with multiple chain pharmacies described as 'big data power houses' (ID15). There were contrasting views regarding the influence of evidence. Most believed that pilot services, trials, local schemes and good practice could lead to change by replication at a national level. However, some community pharmacists did not share this belief.

## DISCUSSION

This study describes the views of a wide variety of systematically selected stakeholders' on how community pharmacy services may be provided in the short term to mid-term, in order to meet current NHS challenges. All types of stakeholder anticipated increased provision of services for long-term conditions, urgent care and public health, and viewed collaboration with general practice, greater use of technology, workforce development, and financial and contractual reforms as facilitators to enable this. Several barriers to these developments were also identified, which were especially apparent in the varying perspectives of internal and external stakeholders. These included an insecure and demotivated workforce, poor relationships between community pharmacy and general practice, and inadequate and disunited leadership and representation.

Expectations that community pharmacists will provide services for long-term conditions, urgent care and public health have been previously reported[19] and are supported by evidence of the effectiveness of such interventions.[4–6] Since the interviews were completed, a number of new nationally commissioned services in these areas have been introduced, including the Discharge Medicines Service, the Community Pharmacist Consultation Service (with GP referrals), the Smoking Cessation Service and the Hypertension Case Finding Service.[20] These services demonstrate the importance of greater collaboration between community pharmacy and general practice, as some require referral of patients between these two settings, and many were initially part of the Pharmacy Integration Fund (PhIF). This was established in 2016 to accelerate the integration of pharmacy services into the wider NHS.[21] The PhIF demonstrates the importance of nationally led piloting and commissioning of pharmacy services, as this overcomes local difficulties in community pharmacy–GP relationships. More recently, the PhIF has funded a community pharmacy clinical lead in all areas of England to further integrate new pharmacy services into the wider NHS. The recent NHS England announcement of investment in community pharmacy-based services to

treat common conditions, provide oral contraception and monitor blood pressure, with the aim of reducing workload in general practice, continues this trend.[22]

However, given the importance of collaboration, actions to overcome the difficult community pharmacy–GP relations described by interviewees will be important, especially as this has been reported before.[19 23] Moreover, this study found for the first time that such competition extends to relationships between community and GP pharmacists, although other intraprofessional tensions, lack of awareness and concern about the financial viability of community pharmacy have been observed previously.[24 25] The growing number of GP pharmacists means that GPs now have greater experience of the benefits of working with the pharmacy profession. This provides an opportunity for individual pharmacists and pharmacy organisations to improve local relationships with GPs, to the benefit of patients and the NHS. For example, interprofessional education is now a requirement in initial training for both pharmacists and doctors and has been shown to improve attitudes towards interprofessional collaboration.[26]

Aspects of workforce development were also key facilitators identified in this study, particularly independent prescribing qualification for pharmacists and extended roles for pharmacy technicians. Although a number of recent initiatives have provided funding for independent prescribing training for community pharmacists, in 2021, there was only one community pharmacist independent prescriber for every 10 community pharmacies, an unchanged proportion from 2017.[27] All newly qualified pharmacists will be independent prescribers from 2026, but further initiatives are required to train the existing workforce.

Better use of automation and technology will be required to use a more skilled workforce efficiently. The NHS currently lacks the national information technology (IT) infrastructure required to give all community pharmacists access to patients' full health records, which was a key facilitator of collaboration with GPs identified in this and other studies.[19] However, a number of local pilot schemes are underway, with a target for national access by March 2025.[28] Interviewees also identified automated dispensing as releasing community pharmacists to provide new services. While there is evidence of the safety, efficiency and cost-effectiveness of automated community pharmacy dispensing,[29] this has not yet been widely adopted by community pharmacies in England.

Financial and contractual reforms to balance income from retail and cognitive activities were identified as important facilitators of future pharmacy services.[30] This is demonstrated by the rapid uptake of newly commissioned national services, such as the Community Pharmacist Consultation Service and the Hypertension Case Finding Service (over 98 000 and 113 000 consultations in October 2022, respectively).[31] The national commissioning of pharmacy services is thus a strong driver for change, particularly as it means patients have access to a consistent range of services from most community pharmacies, encouraging greater use. The PhIF model of national commissioning of services that have undergone a successful centrally funded pilot[21] should, therefore, be continued, to ensure consistent integration of community pharmacy with the wider NHS in all areas of England.

Such developments will only happen with strong leadership, from both the NHS and the community pharmacy sector. Both internal and external stakeholders were clear that community pharmacy leadership and representation is currently poor, particularly because it is provided by multiple organisations that do not present a unified position. Stronger, unified leadership of community pharmacy is, therefore, also required.

## Strengths and limitations

The key strength of this study is its examination of the development of community pharmacy services in England in their entirety (rather than individual services) from the perspective of a wide variety of systematically selected interviewees, making it more representative and generalisable than some previous studies. However, the inability to recruit stakeholders from the government and organisations that represent the public limits the breadth of perspectives included. A separate study examining the perspectives of members of the public has been completed.[18] The addition of pharmacists working in other sectors (eg, GP pharmacists) could have added useful insights. Despite the interviews being conducted in 2018–2019, the results remain salient, if not more so, due to the intervening COVID-19 pandemic which has accelerated development of many of the topics discussed.

## Recommendations

Based on these findings, the Department of Health and Social Care and NHS should implement policies that ensure that community pharmacy is fully integrated with the wider primary care system and can work collaboratively with other healthcare professionals (eg, by enabling full access to health records with appropriate safeguards). Clinical and contractual incentives for the primary care sector should be aligned to avoid creating competitive relationships between professions. In addition, the state should facilitate the use of automated dispensing in community pharmacy, to reduce the amount of community pharmacy staff time used for dispensing services while ensuring access to urgent medicines and face-to-face contact with a pharmacist. The time thus released should then be used to provide new services related to long-term conditions, urgent care, minor ailments and public health, but this may also include reducing health inequalities, providing vaccination and support for people with mental health problems. Commissioning of community pharmacy services should be reformed to create a less fragmented national system with adequate remuneration based on patient-orientated services rather than dispensing volume. Finally, policies are required to develop the skills of the community pharmacy workforce

for both current and anticipated roles, with a particular focus on independent prescribing and pharmacy technicians.

Pharmacy representative organisations should ensure that the community pharmacy sector is represented effectively to multiple audiences (the Government, NHS, other professions and the public) and presents a unified voice and shared vision. They should especially advocate for the implementation of the policies recommended above and increase awareness of the role of community pharmacists among other healthcare professionals and the public. They should also promote a culture of participation in research to generate and publish an evidence base for community pharmacy services that is responsive to NHS needs, as interviewees identified the need for a stronger evidence base for community pharmacist interventions. Finally, community pharmacy businesses and community pharmacists should be proactive in supporting these developments (as interviewees identified resistance to change as one of the biggest barriers to community pharmacy development) and need to take urgent action to support their insecure and demotivated workforce. It should be recognised that some of these recommendations are ambitious and implementation will require careful planning and considerable time.

## CONCLUSION

Stakeholders expect that in the future community pharmacists will help to address current NHS challenges by managing long-term conditions, providing urgent care and public health services. Collaboration with primary care and use of new technologies will be fundamental for facilitating these changes and ensuring they benefit patients. Numerous changes are required from the government, NHS, pharmacy representative bodies, community pharmacy businesses and community pharmacists to create the environment in which these changes can occur. Future research should assess the impact of these changes (especially the effectiveness of new services) and understand more about the evolving professional relationships between community pharmacists, GPs and general practice pharmacists.

**Acknowledgements** The authors would like to sincerely thank all the stakeholders who participated in the study and everyone who helped with their recruitment. The authors would also like to thank Dr Bharat Shah CBE, Sigma Pharmaceuticals and the Harold and Marjorie Moss Charitable Trust Fund. Dr Paloumpi's PhD was funded by The Bharat Shah PhD Scholarship. Dr Paloumpi was also supported by the Harold and Marjorie Moss Charitable Trust Fund.

**Contributors** EP: conceptualisation; methodology; project administration; data curation; formal analysis; writing—original draft. PO: conceptualisation; methodology; supervision; writing—review and editing. MCW: conceptualisation; methodology; supervision; funding acquisition; writing—review and editing. MDJ: conceptualisation; methodology; supervision; funding acquisition; writing—review and editing; guarantor.

**Funding** This work was supported by Sigma Pharmaceuticals via the Bharat Shah PhD Scholarship (award/grant no: NA) and the Harold and Marjorie Moss Charitable Trust Fund (award/grant no: NA).

**Disclaimer** The views expressed in this paper are purely those of the author. They do not necessarily reflect the views or official positions of the European Commission and the ERC Executive Agency.

**Competing interests** None declared.

**Patient and public involvement** Patients and/or the public were not involved in the design, or conduct, or reporting, or dissemination plans of this research.

**Patient consent for publication** Not applicable.

**Ethics approval** This study involves human participants and the Research Ethics Approval Committee for Health (REACH) of University of Bath (reference EP 17/18 204) approved the study in August 2018. Participants gave informed consent to participate in the study before taking part.

**Provenance and peer review** Not commissioned; externally peer reviewed.

**Data availability statement** No data are available.

**ORCID iDs**
Piotr Ozieranski http://orcid.org/0000-0002-2023-3288
Margaret C Watson http://orcid.org/0000-0002-8198-9273
Matthew D Jones http://orcid.org/0000-0002-2617-4098

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
