## [Reviewer comments · BMJ Open]

ARTICLE DETAILS

TITLE (PROVISIONAL)	Professional stakeholders' expectations for the future of community pharmacy practice in England: a qualitative study
AUTHORS	Paloumpi, Evgenia; Ozieranski, Piotr; Watson, Margaret; Jones, Matthew

VERSION 1 – REVIEW

REVIEWER	Hindi, Ali The University of Manchester, Centre for Pharmacy Workforce Studies, Division of Pharmacy and Optometry
REVIEW RETURNED	21-May-2023

GENERAL COMMENTS	Overall comments - This paper addresses an important topic looking at stakeholders' expectations for the future of community pharmacy practice in England. The findings are of great relevance to policymakers. Whilst findings not necessarily novel in that they confirm what is already known, they are of great relevance. It is good to have insights from key stakeholders around issues which are prevalent in community pharmacy.- However, I was surprised to see no patient/public contributors in this study. Also, surprising to see other members of the pharmacy team such as pharmacy technicians not included.- The discussion was well written and demonstrated good awareness of current community pharmacy landscape. However, some of the policy recommendations are very ambitious. Would be good to highlight that implementing that the policy recommendations will be gradual and take a lot of time, considering the challenges this sector faces. Abstract - Not clear how internal and external health-sector stakeholders are defined. Background - "There is evidence of the effectiveness of community pharmacy services to manage long-term conditions⁴, minor ailments⁵ and promote public health" It would be good to mention what kind of evidence (i.e. RCT, Qualitative, Quantitative).- Will be worth mentioning the recently announced £645m investment in community pharmacy and the potential implications moving forward. Methods - Should define internal and external stakeholders- Need to mention how other stakeholders (other than community pharmacists) were identified prior to recruitment.
--

	Results  - Theme 1 - Collaboration for wider pharmacy services provision from community pharmacy: Are specifics around what needs to be done to achieve better collaboration between community pharmacy and the wider healthcare sector. - Any specifics around what needs to be done to foster a community pharmacy culture that promotes collaborative working? - Any explanation as to why some community pharmacists thought they were better placed to support patients compared to GP pharmacists? - Any insights as to whether some/most/all CP described having competitive relationships with GPs. - Theme 2 - integrating technology into pharmacy services: The use of AI in clinical decision seems speculative. I can see how the technical aspects of dispensing could be automated, but not sure if AI can replicate the clinical aspects which require clinical skills and knowledge. - Theme 3 - building workforce competence: "Interviewees referred to the need to train the community pharmacy workforce for such responsibilities, including physical examinations, clinical assessment, consultation skills and data management". Surprised nothing brought up about the pharmacy environment being a huge barrier to providing such services. - "Training will be essential, considering the existing variability in the skills of pharmacy technicians". Please clarify if this was said by stakeholders or this is a general comment by the author - Theme 4: Future community pharmacy services: in terms of long-term conditions, there needs to be more description/consideration of the "difficult journey". There has been previous works which discuss internal/external challenges to provide LTC services such as lack of remuneration, integration, interoperability. I feel this section needs to touch on these points. - Can you elaborate/describe "the community pharmacy network" for the read - Given the importance of remuneration to provision of future CP services, it might be worth having this as a separate theme. - I like the idea of having a figure showing coherence and divergence between stakeholder types. However, I am wondering if its best placed at the end of the results? All throughout the results I was wondering about the coherence and divergence between stakeholder types. Discussion  - Similar to comment made earlier in background section, worth mentioning the recently announced £645m investment in community pharmacy and the potential implications moving forward. - Worth mentioning AMK Hindi study looking at integration of community pharmacy services for patients with long-term conditions as there are a lot of overlap in the findings which support your argument. https://doi.org/10.1186/s12875-019-0912-0 - Would caution against specifically advocating for full access to health records without elaborating on restricting access to other non-healthcare staff. - "In addition, the state should facilitate the use of automated dispensing in community pharmacy, to release pharmacist time for new services". Was not clear what this means. Conclusion  - Any implications for future research?
--	--

REVIEWER	Glass, Beverley James Cook University, Pharmacy
REVIEW RETURNED	09-Jun-2023

GENERAL COMMENTS	This is a well written paper and a pleasure to read - my initial concern was a the age of the data in rapidly moving field - I still believe that the authors have left this data too long before publishing - however they have to a certain extent compensated for this in the discussion but highlighting the new initiatives that have come in since the data were collected - another indication that they have waited too long before publishing. I do however feel that this paper is worthy of publication as it does add value to a limited among of literature in the field. Some corrections to be addressed below:  1. The original Walt and Gilson article for 1994 should be referenced 2. In line 120 examples of non-pharmacy organizations should be included 3. Explain the inclusion of locum status in line 141 4. If the topic guide was developed based on Walt and Gilson I would have expected that this framework would have been used to inform the themes - indicate how the themes relate to this framework in table 1 and in the text 5. In line 729, indicate how the promotion of a culture of research is a recommendation and an outcome from the data 6. Line 731 -733 could benefit from reference to the pharmacists' resistance to change. 7. Good conclusion
---

VERSION 1 – AUTHOR RESPONSE

Comment	Response
Reviewer 1: Dr. Ali Hindi, The University of Manchester Overall comments This paper addresses an important topic looking at stakeholders' expectations for the future of community pharmacy practice in England. The findings are of great relevance to policymakers. Whilst findings not necessarily novel in that they confirm what is already known, they are of great relevance. It is good to have insights from key stakeholders around issues which are prevalent in community pharmacy.	No response required.
However, I was surprised to see no patient/public contributors in this study. Also, surprising to see other members of the pharmacy team such as pharmacy technicians not included.	As explained in Section 2.1, representatives from a number of patient organisations were invited to participate, but did not respond. This limitation is subsequently discussed in Section 4.1, which also explains that a separate study examining the perspectives of members of the public was completed. We chose to publish this separately in order to give sufficient space to consider the views of all types of stakeholder, and this paper is currently under review. In addition, a representative of the Association of Pharmacy Technicians UK was interviewed for this study (Table S2)

The discussion was well written and demonstrated good awareness of current community pharmacy landscape. However, some of the policy recommendations are very ambitious. Would be good to highlight that implementing that the policy recommendations will be gradual and take a lot of time, considering the challenges this sector faces.	Thank you for this useful suggestion. We have added the following at the end of Section 4.2: “It should be recognised that some of these recommendations are ambitious and implementation will require careful planning and considerable time.”
Abstract	
Not clear how internal and external health-sector stakeholders are defined.	The ‘Participants’ section now reads: “External stakeholders were representatives of non-pharmacy organisations, including policymakers, commissioners and representatives of healthcare professions. Internal stakeholders were community pharmacists or pharmacy organisations representatives.” Other minor changes made to the abstract, to ensure it meets the 300-word limit.
Background	
“There is evidence of the effectiveness of community pharmacy services to manage long-term conditions, minor ailments and promote public health” It would be good to mention what kind of evidence (i.e. RCT, Qualitative, Quantitative).	We have added “...from systematic reviews and quantitative primary research.” to the end of this sentence (lines 84-85).
Will be worth mentioning the recently announced £645m investment in community pharmacy and the potential implications moving forward.	Thank you for highlighting this important announcement, which was made after we submitted our manuscript. We have now included this in the ‘Discussion’ in lines 653-655: “The recent NHS England announcement of investment in community pharmacy-based services to treat common conditions, provide oral contraception and monitor blood pressure, with the aim of reducing workload in general practice, continues this trend.” We have not included it in the ‘Introduction’ section, as this development occurred after the research was carried out, so it fits better in the ‘Discussion’ section.
Methods	
Should define internal and external stakeholders	These are defined in Section 2.1: “External stakeholders were representatives of non-pharmacy organisations. Internal stakeholders were either current community pharmacists or representatives of pharmacy organisations.”
Need to mention how other stakeholders (other than community pharmacists) were identified prior to recruitment.	This is explained by the description of the stakeholder mapping process in paragraph 2 of Section 2.1. To make it clear that this paragraph is describing the recruitment of internal and external organisational representatives, we have added “...stakeholder mapping and...” to line 124 and “...internal and external...” to lines 127-128.
Results	
Theme 1 - Collaboration for wider pharmacy services provision from community pharmacy: Are specifics around what needs to be done to achieve better collaboration between community	Participants did not say much about how collaboration could be improved. The most relevant topics discussed are included in the manuscript: collaboration with primary care

pharmacy and the wider healthcare sector.	networks (lines 217-219) and a culture change (lines 221-223). In addition, Section 4.1 contains a recommendation to align contractual incentives in primary care to avoid creating competitive relationships (lines 721-723).
Any specifics around what needs to be done to foster a community pharmacy culture that promotes collaborative working?	This is linked to the above comment and was not discussed in detail by participants.
Any explanation as to why some community pharmacists thought they were better placed to support patients compared to GP pharmacists?	This is summarised by the quotation from participant ID14 (lines 227-231). We have added additional description in lines 240-241: "...due to better practical knowledge of patients' current medicines and how they are using them."
Any insights as to whether some/most/all CP described having competitive relationships with GPs.	This was the view of most community pharmacists interviewed. We have clarified this by adding "Many..." to the start of line 248.
Theme 2 - integrating technology into pharmacy services: The use of AI in clinical decision seems speculative. I can see how the technical aspects of dispensing could be automated, but not sure if AI can replicate the clinical aspects which require clinical skills and knowledge.	It's fair to say that participants' discussion of AI was speculative, but this was brought up by many external stakeholders. To highlight this, we have added "...discussed speculatively" to line 284.
Theme 3 - building workforce competence: "Interviewees referred to the need to train the community pharmacy workforce for such responsibilities, including physical examinations, clinical assessment, consultation skills and data management". Surprised nothing brought up about the pharmacy environment being a huge barrier to providing such services.	Community pharmacists described several barriers to training, "...including time, funding and accessibility" (lines 399-401). Time pressure is related to the pharmacy environment, but participants did not discuss the physical environment in relation to training.
"Training will be essential, considering the existing variability in the skills of pharmacy technicians". Please clarify if this was said by stakeholders or this is a general comment by the author	This was discussed by community pharmacists, it is not a comment from the authors. We have clarified this in lines 429-430: "...described by some community pharmacists."
Theme 4: Future community pharmacy services: in terms of long-term conditions, there needs to be more description/consideration of the "difficult journey". There has been previous works which discuss internal/external challenges to provide LTC services such as lack of remuneration, integration, interoperability. I feel this section needs to touch on these points.	The challenges associated with services for long-term conditions were only discussed by a minority of participants, but we have expanded on this section as follows (lines 456-460): "...that is difficult for pharmacists who are "tied to a pharmacy" (interviewee ID9 – internal stakeholder) with limited capacity to take on additional roles and access training. This sub-theme therefore links to others related to collaboration, the potential of technology to change pharmacy practice and workforce competence."
Can you elaborate/describe "the community pharmacy network" for the read	This was just intended to refer to the approximately 11,000 community pharmacies in England. We have revised it to read "Community pharmacies..." to avoid this confusion (line 489).
Given the importance of remuneration to provision of future CP services, it might be worth having this as a separate theme.	We considered this during the analysis process. As participants regularly linked remuneration to commissioning, representation and evidence, we decided that the existing thematic structure was a better representation of the data.
I like the idea of having a figure showing coherence and divergence between stakeholder types. However, I am wondering if its best placed	Thank you for this suggestion. We have moved this figure and the associated discussion to the end of Section 3.2.

at the end of the results? All throughout the results I was wondering about the coherence and divergence between stakeholder types.	
Discussion	
Similar to comment made earlier in background section, worth mentioning the recently announced £645m investment in community pharmacy and the potential implications moving forward.	As described above, we have now included this in lines 653-655: “The recent NHS England announcement of investment in community pharmacy-based services to treat common conditions, provide oral contraception and monitor blood pressure, with the aim of reducing workload in general practice, continues this trend.”
Worth mentioning AMK Hindi study looking at integration of community pharmacy services for patients with long-term conditions as there are a lot of overlap in the findings which support your argument. https://doi.org/10.1186/s12875-019-0912-0	Thank you for highlighting the overlapping findings of this paper. We have now referenced it in lines 640, 659 and 682.
Would caution against specifically advocating for full access to health records without elaborating on restricting access to other non-healthcare staff.	Interestingly, this is discussed in more detail in our other paper (under review) presenting patients’ perspectives on community pharmacy. We have added “...with appropriate safeguards...” in line 721.
“In addition, the state should facilitate the use of automated dispensing in community pharmacy, to release pharmacist time for new services”. Was not clear what this means.	We have revised this to read (lines 723-727): “In addition, the state should facilitate the use of automated dispensing in community pharmacy, to reduce the amount of community pharmacy staff time used for dispensing services. The time thus released should then be used to provide new services related to long-term conditions, urgent care, minor ailments and public health...”
Conclusion	
Any implications for future research?	We have added some recommendations in lines 754-756: “Future research should assess the impact of these changes (especially the effectiveness of new services) and understand more about the evolving professional relationships between community pharmacists, GPs and general practice pharmacists.”
Reviewer 2: Dr. Beverley Glass, James Cook University	
This is a well written paper and a pleasure to read - my initial concern was a the age of the data in rapidly moving field - I still believe that the authors have left this data too long before publishing - however they have to a certain extent compensated for this in the discussion but highlighting the new initiatives that have come in since the data were collected - another indication that they have waited too long before publishing. I do however feel that this paper is worthy of publication as it does add value to a limited among of literature in the field. Some corrections to be addressed below:	No response required.
1. The original Walt and Gilson article for 1994 should be referenced	We have added this reference (line 154).

2. In line 120 examples of non-pharmacy organizations should be included	We have added a new Table S2 to the supplementary information, which lists all the organisations approached to participate. This is referred to in the main manuscript in lines 121 and 130-131. The organisations that participated were already listed in what is now Table S3.
3. Explain the inclusion of locum status in line 141	We have added the following explanation in lines 143-144: “...(as these pharmacists work in multiple pharmacies and so may have different perspectives)...”
4. If the topic guide was developed based on Walt and Gilson I would have expected that this framework would have been used to inform the themes - indicate how the themes relate to this framework in table 1 and in the text	The topic guide was developed using a recent policy review that employed the Walt and Gilson framework (lines 152-154). It was the policy review findings, not Walt and Gilson, that informed the topic guide. The Walt and Gilson framework was in effect one step removed from the topic guide. This explains why the constructs of the Walt and Gilson framework are not seen in the themes. The themes were generated inductively (line 160) and so their structure is based on the data.
5. In line 729, indicate how the promotion of a culture of research is a recommendation and an outcome from the data	We have clarified this with: “...as interviewees identified the need for a stronger evidence base for community pharmacist interventions.” (Lines 742-743).
6. Line 731 -733 could benefit from reference to the pharmacists' resistance to change.	We have added “...(as interviewees identified resistance to change as one of the biggest barriers to community pharmacy development)...” to lines 744-746.
7. Good conclusion	No response required.

VERSION 2 – REVIEW

REVIEWER	Hindi, Ali The University of Manchester, Centre for Pharmacy Workforce Studies, Division of Pharmacy and Optometry
REVIEW RETURNED	28-Jul-2023
GENERAL COMMENTS	Many thanks for addressing the comments thoroughly. I want to thank the authors for an enjoyable read. I have no further comments.